# Low-Finesse Fabry–Pérot Interferometers Applied in the Study of the Relation between the Optical Path Difference and Poles Location

**DOI:** 10.3390/s20020453

**Published:** 2020-01-13

**Authors:** José Trinidad Guillen Bonilla, Héctor Guillen Bonilla, Verónica María Rodríguez Betancourtt, María Eugenia Sánchez Morales, Juan Reyes Gómez, Antonio Casillas Zamora, Alex Guillen Bonilla

**Affiliations:** 1Departamento de Electrónica, Centro Universitario de Ciencias Exactas e Ingenierías (C.U.C.E.I.), Universidad de Guadalajara, Blvd. M. García Barragán 1421, Guadalajara 44410, Mexico; trinidad.guillen@academicos.udg.mx; 2Departamento de Matemáticas, Centro Universitario de Ciencias Exactas e Ingenierías (C.U.C.E.I.), Universidad de Guadalajara, Blvd. M. García Barragán 1421, Guadalajara 44410, Mexico; 3Departamento de Ingeniería de Proyectos, Centro Universitario de Ciencias Exactas e Ingenierías (C.U.C.E.I.), Universidad de Guadalajara, Blvd. M. García Barragán 1421, Guadalajara 44410, Mexico; hector.guillen@academicos.udg.mx (H.G.B.); antonio.czamora@academicos.udg.mx (A.C.Z.); 4Departamento de Química, Centro Universitario de Ciencias Exactas e Ingenierías (C.U.C.E.I.), Universidad de Guadalajara, Blvd. M. García Barragán 1421, Guadalajara 44410, Mexico; veronica.rbetancourtt@academicos.udg.mx; 5Departamento de Ciencias Tecnológicas, Centro Universitario de la Ciénega (CUCienéga), Universidad de Guadalajara, Av. Universidad No. 1115, LindaVista, C.P., Ocotlán 47810, Mexico; eugenia.sanchez@cuci.udg.mx; 6Departamento de Ciencias químicas, Universidad de Colima, Las Víboras, Coquimatlan 28045, Mexico; reyesgj@ucol.mx; 7Departamento de Ciencias Computacionales e Ingenierías, Centro Universitario de los Valles (CUValles), Universidad de Guadalajara, Carretera Guadalajara-Ameca Km. 45.5, Ameca 46600, Mexico

**Keywords:** relation between the optical path difference (OPD) and poles location, interferometry sensors, Fabry–Pérot interferometer, Laplace transform, pole-zero map

## Abstract

Interferometry sensors are frequently analyzed by applying the Fourier transform because the transformation separates all frequency components of its signal, making its study on a complex plane feasible. In this work, we study the relation between the optical path difference (OPD) and poles location theoretically and experimentally, using the Laplace transform and a pole-zero map. Theory and experiments are in concordance. For our study, only the cosine function was considered, which is filtered from the interference pattern. In experimental work, two unperturbed low-finesse Fabry–Pérot interferometers were used. First, a Fabry–Pérot interferometer that has a cavity length of ~1.6 mm was used. Its optical path difference was 2.33 mm and the poles were localized at points ±i12. rad/nm. Secondly, a Fabry–Pérot interferometer with a cavity length of ~5.2 mm was used, and its optical path difference was 7.59 mm and the poles were localized at points ±i40.4 rad/nm. Experimental results confirmed the theoretical analysis. Our proposal finds practical application for interferometer analysis, signal processing of optical fiber sensors, communication system analysis, and multiplexing systems based on interferometers.

## 1. Introduction

Many implemented interferometers are based on Bragg gratings, optic fiber, mirrors, crystals, and their combinations [1,2,3,4,5]. Interferometers find practical applications for physical parameter measurements, such as temperature, strain, humidity, pressure, level, current, voltage, and vibration [6,7,8,9,10]. When used for measurement, any interferometer suffers the modification of its optical path difference (OPD) due to external perturbations where the OPD parameter is the relative path difference (or phase shift) traveled between two light beams emitted from two coherent sources. Thus, the optical path difference measurement plays a very important role for the interferometry system application. Diverse transformations were applied to study the OPD parameter, for example: Fourier transform and Hilbert transform [11,12,13,14]. Their studies permitted the signal demodulation for the interferometry systems, and, consequently, new sensing systems were developed [15,16].

Graphical methods are frequently applied to study the dynamic behavior of any continuous system. Some methods are pole-zero maps, bode diagrams, polar, and Nichols diagrams [17,18,19,20]. Each graphical representation has its own characteristics, but all graphical techniques frequently require a complex s-function. In particular, a pole-zero map shows graphically pole-zeros in the s-complex plane. Poles and zeros have been calculated from a complex function which was determined by applying the Laplace transform to the system’s equation. Necessarily, this complex function contains the complete information about the physical parameter under study, since otherwise the system wouldn’t be correctly studied.

On the other hand, in the dynamic system analysis, the complex s-function $F\left(s\right)=\frac{P\left(s\right)}{G\left(s\right)}$ describes the system under study completely, where P(s) and G(s) are complex polynomials in terms of s=iω+σ: i is the imaginary operator, ω is the angular frequency, and σ is a real parameter. Poles are calculated through the polynomial G(s), and, using the polynomial P(s), the zeros are calculated. Both poles and zeros can be graphed in the pole-zero map representation [18,19]. To our knowledge, the dependence of the pole location due to the optical path difference has not been reported in the literature. If this study is developed, an interferometric system based on two beams can be analyzed on the s-plane, making it feasible to apply graphical methods to study the dynamic behavior of interferometers, and as a consequence, the graphical methods can simplify the signal demodulation of interferometer systems. In this work, using a pole-zero map and two low-finesse Fabry–Pérot interferometers printed inside the fiber’s core, the behavior of optical path difference vs. pole location was theoretically and experimentally corroborated. Theory and experiments are in concordance. To study the relation between both terms, all frequency components were calculated from the optical signal produced by the two interferometers. Using two filters, the cosine functions from both interference patterns were filtered and then its inverse Fourier transform was calculated for each interference pattern. Following this, the Laplace transform was applied to obtain a complex function, which was used to calculate the poles and zeros, making it feasible to build a pole-zero map. Based on our results, when the interferometers have their cavities of 1.6 mm and 5.2 mm, the optical path differences are 2.36 mm and 7.63 mm, while the pole locations are $\pm i12.50\;\frac{rad}{nm}$ and $\pm i40.58\;\frac{rad}{nm}$, respectively. Poles are always over the imaginary axis and a zero is over the origin. Our analysis considers only the cosine function of an interference pattern produced between two beams, and the interferometer doesn’t have external perturbations.

## 2. Materials and Methods

### 2.1. Optical System

Figure 1 shows the optical system under study. The system consisted of a single-mode optical fiber (FS-PM-7621). Its characteristics were: refraction index 1.46, operating wavelength 1550 nm, nominal numerical aperture 0.13, attenuation $\alpha =1\frac{Db}{Km}$, cladding diameter 125 μm, core diameter 9.5 μm, and length L=4 m. The fiber had two low finesse Fabry–Pérot interferometers printed inside the fiber’s core. Both Fabry–Pérot interferometers were formed by two identical Bragg gratings. The length of cavities were $L_{FP1}\approx 1.6\;mm$ and $L_{FP2}\approx 5.2\;mm$ and the separation, $L_{SR}\approx 1.2\;m$ (between both interferometers), eliminated ghost interferometers. All Bragg gratings have a rectangular profile, with a Bragg wavelength at $\lambda_{BG}\approx 1535\;nm$, a length at $L_{BG}\approx 0.5\;mm$, and low reflectivity (≈1%), eliminating cross-talk noise. The fiber optic circulator (6015-3-FC) couples a beam light between the optical broadband source (Thor-Labs model SLD1005S) and the single-mode optical fiber. Finally, the reflection spectrum generated from both interferometers was detected using the optical analyzer spectrometer (Q8384 OSA spectrometer, brand ADVANTEST).

### 2.2. Optical Signal

Based on References [21,22], when both interferometers do not have external perturbations, the optical signal $R_{T}\left(\lambda \right)$ is the superposition of two interference patterns, which are generated from both Fabry–Pérot interferometers. Its mathematical representation can be written as:(1)$$R_{T}\left(\lambda \right)=\sum_{m=1}^{2}2a_{m}\left\{b^{2}\sin c^{2}\left(\frac{\lambda -\lambda_{BG}}{\Delta_{BG}}\right)\right\}\left\{1+\cos\left[2\pi \nu_{FPm}\left(\lambda -\lambda_{BG}\right)\right]\right\},$$
where $a_{m}$ is an amplitude factor and the term b is given by
(2)$$b=\frac{\pi n_{1}L_{BG}}{\lambda_{BG}},$$
where the width $\Delta_{BG}$ is the spectral distance between its +1 and −1 zeros,
(3)$$\Delta_{BG}=\frac{\lambda_{BG}^{2}}{n_{1}L_{BG}},$$
and the frequency component $\nu_{FPm}$ is defined as
(4)$$\nu_{FPm}=\frac{2nL_{FMm}}{\lambda_{BG}^{2}}\;m=1,\;2,$$
where λ is the wavelength, $n_{1}$ is the the amplitude of the effective refraction index modulation of the gratings, and $L_{FMm}$ is the *m-th* cavity length (*m* = 1, 2). From Equation (1), each interference pattern is formed by an enveloped function and a modulate function. The enveloped function is the sinc^2^(x) and the modulate function is a constant plus the cosine function.

Applying the Fourier transform, the frequency components can be calculated, and, as a consequence, these can be graphed on the complex plane. Therefore, we applied the Fourier transform to the optical signal.
(5)$$R_{T}\left(\nu \right)=\int_{-\infty }^{\infty }R_{T}\left(\lambda \right)e^{-i2\pi \nu \lambda }d\lambda ,$$
where $R_{T}\left(\nu \right)$ is the frequency spectrum and ν is the frequency. Substituting Equation (1) into Equation (5), the frequency spectrum is
(6)$$R_{T}\left(\nu \right)=\int_{-\infty }^{\infty }\sum_{m=1}^{2}2a_{m}\left\{b^{2}sinc^{2}\left(\frac{\lambda -\lambda_{BG}}{\Delta_{BG}}\right)\right\}\left\{1+cos\left[2\pi \nu_{FPm}\left(\lambda -\lambda_{BG}\right)\right]\right\}e^{-i2\pi \nu \lambda }d\lambda .$$

Invoking the Fourier transform properties and series properties, $R_{T}\left(\nu \right)$ can be expressed through
(7)$$R_{T}\left(\nu \right)=\sum_{m=1}^{2}ℱ\left\{2a_{m}\left\{b^{2}sinc^{2}\left(\frac{\lambda -\lambda_{BG}}{\Delta_{BG}}\right)\right\}\right\}⊗ℱ\left\{1+cos\left[2\pi \nu_{FPm}\left(\lambda -\lambda_{BG}\right)\right]\right\},$$
where the symbol ℱ{ } is the Fourier operator and the symbol ⊗ indicates the convolution operation. Invoking the convolution properties, using the identities $\cos\left(\varphi \right)=\frac{e^{i\varphi }+e^{-i\varphi }}{2}$, $\cos^{2}\left(\varphi \right)=\frac{1}{2}\left(1+\cos\left(2\varphi \right)\right)$, and $\sum_{m=1}^{M}e^{-i\varphi }=\sum_{m=-M}^{-1}e^{i\varphi }$ and solving the transformation, the optical signal $R_{T}\left(\nu \right)$ takes the form.
(8)$$R_{T}\left(\nu \right)=\sum_{m=-2}^{2}c_{m}tri\left(\frac{\nu -\nu_{FPm}}{\nu_{BG}}\right)$$

$R_{T}\left(\nu \right)$ is formed by five triangle functions, where the triangle function tri(x) is defined as $tri\left(x\right)=\{\begin{array}{cc} 1-\left|x\right| & \left|x\right|\leq 1\\ 0 & \mathrm{otherwise} \end{array}$, $c_{m}$ are amplitudes factors and $\nu_{BG}$ is the bandwidth.
(9)$$\nu_{BG}=\frac{4n_{1}L_{BG}}{\lambda_{BG}^{2}}$$

Figure 2 shows the frequency spectrum $R_{T}\left(\nu \right)$.

Analyzing Figure 2, the direct component $\left(\nu_{FP0}\right)$ contained information from both Fabry–Pérot interferometers; the components $\pm \nu_{FP1}$ contained information from the first interferometer and the components $\pm \nu_{FP2}$ contained information from our second interferometer. Four lateral components contained information about the cosine function and the enveloped function, such that, the cosine functions could be filtered from the lateral components, as we describe in the next section.

### 2.3. Cosine Function Determination

To filter both cosine functions generated from both Fabry–Pérot interferometers, the following system was considered.
(10)$$F_{m}\left(\nu \right)=R_{T}\left(\nu \right)T_{m}\left(\nu \right)\;m=1,2,$$
where $T_{m}\left(\nu \right)$ is the m-th filter and $F_{m}\left(\nu \right)$ is the m-th cosine function. The first filter $T_{1}\left(\nu \right)$ consisted of two unitary Dirac deltas, which were centered at $\pm \nu_{FP1}$,
(11)$$T_{1}\left(\nu \right)=\delta \left(\nu -\nu_{FP1}\right)+\delta \left(\nu +\nu_{FP1}\right),$$
and our second filter $T_{2}\left(\nu \right)$ consisted of two unitary Dirac delta, but their locations were $\pm \nu_{FP2}$
(12)$$T_{2}\left(\nu \right)=\delta \left(\nu -\nu_{FP2}\right)+\delta \left(\nu +\nu_{FP2}\right).$$

Developing the operation described in Equation (10), both cosine functions were filtered from the optical signal $R_{T}\left(\nu \right)$, and its graphical representation is shown in Figure 3.

Based on Figure 3, the enveloped function information was eliminated, but the cosine function was conserved. We can then study the relationship between optical path difference and poles location. Following this, the inverse Fourier transform was applied to the signal $F_{m}\left(\nu \right)$,
(13)$$f_{m}\left(\lambda \right)=\int_{-\infty }^{\infty }F_{m}\left(\nu \right)e^{i2\pi \nu \lambda }d\nu ,$$
and then we obtained both cosine functions in the wavelength domain
(14)$$f_{m}\left(\lambda \right)=2a_{1}b^{2}cos\left(2\pi \nu_{FP1}\lambda \right)+2a_{2}b^{2}cos\left(2\pi \nu_{FP2}\lambda \right).$$

Considering Equation (4) and the next condition $a_{1}\approx a_{2}\approx a$, Expression (14) takes the form,
(15)$$f_{m}\left(\lambda \right)=2ab^{2}cos\left(\frac{4\pi nL_{FP1}}{\lambda_{BG}^{2}}\lambda \right)+2ab^{2}cos\left(\frac{4\pi nL_{FP2}}{\lambda_{BG}^{2}}\lambda \right).$$

The condition $a_{1}\approx a_{2}\approx a$ is acceptable because all Bragg gratings have the same characteristics. Let us introduce the optical path difference definition $\left(OPD_{m}=nL_{FPm}\right)$ for our last expression, obtaining the next Equation,
(16)$$f_{m}\left(\lambda \right)=f_{1}\left(\lambda \right)+f_{2}\left(\lambda \right)=2ab^{2}cos\left(\frac{4\pi OPD_{1}}{\lambda_{BG}^{2}}\lambda \right)+2ab^{2}cos\left(\frac{4\pi OPD_{2}}{\lambda_{BG}^{2}}\lambda \right).$$

Figure 4 shows both cosine functions in the wavelength domain.

From Equation (16) and Figure 4, each Fabry–Pérot interferometer has its own optical path difference because each interferometer has its own cavity length, and, as a consequence, each cosine function has its own frequency. These frequecies can be vizualized through a pole-zero-map. Notice that cosine’s amplitude depends of physical parameters as a, Bragg wavelength $\lambda_{BG}$, the amplitude of the effective refraction index modulation of the gratings $n_{1}$, and the length of Bragg grating $L_{BG}$ (See Equation (2)).

### 2.4. Pole-Zero Map Representation

To graph the pole-zero map, the unilateral Laplace transform was calculated for each cosine function,
(17)$$F_{m}\left(s\right)=\int_{-\infty }^{\infty }f_{m}\left(\lambda \right)e^{-s\lambda }d\lambda =\int_{-\infty }^{\infty }f_{1}\left(\lambda \right)e^{-s\lambda }d\lambda +\int_{-\infty }^{\infty }f_{2}\left(\lambda \right)e^{-s\lambda }d\lambda .$$

Substituting Equation (16) into Equation (17),
(18)$$F_{m}\left(s\right)=2ab^{2}\int_{-\infty }^{\infty }cos\left(\frac{4\pi OPD_{1}}{\lambda_{BG}^{2}}\lambda \right)e^{-s\lambda }d\lambda +2ab^{2}\int_{-\infty }^{\infty }cos\left(\frac{4\pi OPD_{2}}{\lambda_{BG}^{2}}\lambda \right)e^{-s\lambda }d\lambda ,$$
and solving the transformation, the complex function $F_{m}\left(s\right)$ takes the form,
(19)$$F_{m}\left(s\right)=F_{1}\left(s\right)+F_{2}\left(s\right)=\frac{2ab^{2}s}{s^{2}+{\left(\frac{4\pi OPD_{1}}{\lambda_{BG}^{2}}\right)}^{2}}+\frac{2ab^{2}s}{s^{2}+{\left(\frac{4\pi OPD_{2}}{\lambda_{BG}^{2}}\right)}^{2}}.$$

Substituting Equation (2) into Equation (19), the complex function $F_{m}\left(s\right)$ is
(20)$$F_{m}\left(s\right)=F_{1}\left(s\right)+F_{2}\left(s\right)=\frac{2a{\left(\frac{\pi n_{1}L_{BG}}{\lambda_{BG}}\right)}^{2}s}{s^{2}+{\left(\frac{4\pi OPD_{1}}{\lambda_{BG}^{2}}\right)}^{2}}+\frac{2a{\left(\frac{\pi n_{1}L_{BG}}{\lambda_{BG}}\right)}^{2}s}{s^{2}+{\left(\frac{4\pi OPD_{2}}{\lambda_{BG}^{2}}\right)}^{2}}.$$

Observing Equation (20), the complex function $F_{m}\left(s\right)$ contains information on the optical path differences (OPDs). $F_{1}\left(s\right)$ contains information about the first Fabry–Pérot interferometer and $F_{2}\left(s\right)$ contains information about the second Fabry–Pérot interferometer. From Equation (20), the complex function $F_{1}\left(s\right)$ is defined as
(21)$$F_{1}\left(s\right)=\frac{2a{\left(\frac{\pi n_{1}L_{BG}}{\lambda_{BG}}\right)}^{2}s}{s^{2}+{\left(\frac{4\pi OPD_{1}}{\lambda_{BG}^{2}}\right)}^{2}}.$$

Its zero is localized at the origin because $2a{\left(\frac{\pi n_{1}L_{BG}}{\lambda_{BG}}\right)}^{2}s_{1}=0\to s_{1}=0$, and its poles’ location is
(22)$$s^{2}+{\left(\frac{4\pi OPD_{1}}{\lambda_{BG}^{2}}\right)}^{2}=0\;\to \;\begin{array}{c} s_{1}=i\frac{4\pi OPD_{1}}{\lambda_{BG}^{2}}\\ s_{2}=-i\frac{4\pi OPD_{1}}{\lambda_{BG}^{2}} \end{array}\;.$$

Figure 5a shows the pole-zero map representation. Following this, the complex function $F_{2}\left(s\right)$ is:(23)$$F_{2}\left(s\right)=\frac{2a{\left(\frac{\pi n_{1}L_{BG}}{\lambda_{BG}}\right)}^{2}s}{s^{2}+{\left(\frac{4\pi OPD_{2}}{\lambda_{BG}^{2}}\right)}^{2}}.$$

Again, its zero is localized at the origin, $2a{\left(\frac{\pi n_{1}L_{BG}}{\lambda_{BG}}\right)}^{2}s_{1}=0$. However, the poles are localized at
(24)$$s^{2}+{\left(\frac{4\pi OPD_{2}}{\lambda_{BG}^{2}}\right)}^{2}=0\;\to \;\begin{array}{c} s_{1}=i\frac{4\pi OPD_{2}}{\lambda_{BG}^{2}}\\ s_{2}=-i\frac{4\pi OPD_{2}}{\lambda_{BG}^{2}} \end{array}.$$

Figure 5b shows the pole-zero map for our optical system.

Based on Figure 5, both zeros are over the origin, but the pole locations depend on the optical path difference (OPD). Thus, in the pole-zero map representation, each interference pattern generates its own pole location.

## 3. Results

### 3.1. Optical Signals

To detect the optical signal $R_{T}\left(\lambda \right)$, the OSA spectrometer parameters were: a spectrometer resolution of Δλ=20 nm, the number of samples N = 1000, and a selected dynamic range between 1533 and 1536 nm. The optical broadband source parameters were: power 16 mW and a dynamic range between 1510 and 1570 nm. Finally, Figure 1 shows the sensor parameters. The detected optical signal can be observed in Figure 6a. The measured bandwidth was $\Delta_{BG}\approx 3.18\;nm$ (the theoretical value was $\Delta_{BG}=3.22\;nm$) and the signal-to-noise ratio was SNR = 174.24. Following this, the frequency spectrum $R_{T}\left(\nu \right)$ was calculated and five frequency components were obtained, as Figure 6b illustrates. Their central frequencies were $\nu_{FP0}=0$, $\pm \nu_{FP1}\approx 1.91\;nm^{-1}$, and $\pm \nu_{FP2}\approx 6.43\;nm^{-1}$. Five peaks had the same bandwidth, $\nu_{BG}\approx 1.32\;nm^{-1}$ (the theoretical value was $\nu_{BG}=1.23\;nm^{-1}$).

### 3.2. Cosine Functions

Applying the system described by Equation (10) and two filters defined as
(25)$$T_{1}\left(\nu \right)=\delta \left(\nu -1.91\right)+\delta \left(\nu +1.91\right)$$
and
(26)$$T_{2}\left(\nu \right)=\delta \left(\nu -6.43\right)+\delta \left(\nu +6.43\right),$$
both cosine functions were filtered from the frequency spectrum $R_{T}\left(\nu \right)$, see Figure 7a,b. Calculating its inverse Fourier transform, the cosine functions were represented in the wavelength domain, as Figure 7c,d illustrates. Based on Figure 7, the first cosine function had the frequency of $\nu_{FP1}\approx 1.91\;nm^{-1}$ and its amplitude was $2a{\left(\frac{\pi n_{1}L_{BG}}{\lambda_{BG}}\right)}^{2}\approx 5.7\times 10^{-11}$; the second cosine function had a frequency of $\nu_{FP2}\approx 6.43\;nm^{-1}$ and its amplitude was $2a{\left(\frac{\pi n_{1}L_{BG}}{\lambda_{BG}}\right)}^{2}\approx 4.48\times 10^{-11}$. Considering Equation (14), the mathematical representation is
(27)$$f_{m}\left(\lambda \right)=f_{1}\left(\lambda \right)+f_{2}\left(\lambda \right)=5.7cos\left[2\pi \left(1.91\right)\lambda \right]\times 10^{-11}\left[nm\right]+4.48cos\left[2\pi \left(6.43\right)\lambda \right]\;\times 10^{-11}\left[nm\right]$$
where
(28)$$f_{1}\left(\lambda \right)=5.7cos\left[2\pi \left(1.91\right)\lambda \right]\times 10^{-11}\;\left[nm\right]$$
and
(29)$$f_{2}\left(\lambda \right)=4.48cos\left[2\pi \left(6.43\right)\lambda \right]\;\times 10^{-11}\;\left[nm\right].$$

Their optical path differences were estimated as
(30)$$\begin{array}{c} OPD_{1}=nL_{FP1}=2.33\;mm\\ OPD_{2}=nL_{FP2}=7.59\;mm \end{array}.$$

### 3.3. Pole-Zero Map Representation

Using the cosine functions obtained in Section 3.2, based on Section 2.4 and knowing that our goal verifies the relation between optical path difference and pole location, the unilateral Laplace transform was calculated for both cosine functions obtained experimentally,
(31)$$\begin{array}{c} F_{1}\left(s\right)=5.7\times 10^{-11}\cdot \int_{0}^{\infty }cos\left[2\pi \left(1.91\right)\lambda \right]e^{-s\lambda }d\lambda \\ F_{2}\left(s\right)=4.48\times 10^{-11}\cdot \int_{0}^{\infty }cos\left[2\pi \left(6.43\right)\lambda \right]e^{-s\lambda }d\lambda . \end{array}$$

Solving the transformations, we obtained the next complex functions.
(32)$$\begin{array}{c} F_{1}\left(s\right)=\frac{5.7\times 10^{-11}\;s}{s^{2}+{\left[2\pi \left(1.91\right)\right]}^{2}}\\ F_{2}\left(s\right)=\frac{4.48\times 10^{-11}\;s}{s^{2}+{\left[2\pi \left(6.43\right)\right]}^{2\;.}} \end{array}$$

From Equation (32), $F_{1}\left(s\right)$ has one zero over the origin and two poles localized at points
(33)$$\begin{array}{c} s_{1}=i2\pi \left(1.91\right)=i12\\ s_{2}=-i2\pi \left(1.91\right)=-i12, \end{array}$$
where as $F_{2}\left(s\right)$ has one zero over the origin but their poles are localized at the points.
(34)$$\begin{array}{c} s_{1}=i2\pi \left(6.43\right)=i40.4\\ s_{2}=-i2\pi \left(6.43\right)=-i40.4. \end{array}$$

Figure 8 shows the pole-zero map representation.

## 4. Discussion

In this work, Fabry–Pérot interferometers are studied theoretically and experimentally on the s-plane. Both theory and experiments are in concordance. Such concordance can be observed in Figure 2, Figure 3, Figure 4, Figure 5, Figure 6, Figure 7 and Figure 8, where Figure 2, Figure 3, Figure 4 and Figure 5 show the theoretical results and Figure 6, Figure 7 and Figure 8 show the experimental results. From our theoretical analysis and experimental results, each Fabry–Pérot interferometer has its own frequency channel and the cosine function can be filtered from each interference pattern. Because the cosine function contains the optical path difference information, using this function, we studied the relation between the optical path difference and pole location, as Figure 8 illustrates. Based on Figure 8, each Fabry–Pérot interferometer produced an interference pattern, where each interference pattern had its own frequency and then both frequencies were visualized in the pole-zero map.

From our theoretical analysis and experimental results, it is possible to infer a few key points:(a)Interferometry systems can be studied on the complex s-plane;(b)The modulated function can be expressed as an s-complex function F(s), applying the Laplace transform;(c)The cosine function filtered from the interference pattern always has one zero $s_{1}$ and two poles $s_{1},\;s_{2}$;(d)The zero $s_{1}$ is over the origin and it contains the amplitude information;(e)Both poles $s_{1},\;s_{2}$ are over the imaginary axes. The frequency defines the poles location, and, as a consequence, their locations depend on the optical path difference; see Figure 5 and Figure 7;(f)The pole-zero map gives us information about the optical path difference (OPD);(g)Physical parameters are measured on the complex s-plane;(h)Theoretical and experimental results have small variations due to numerical errors and variations between theoretical and experimental parameters;(i)Since the fiber FS-PM-7621 is polarization maintaining, the polarization effects do not affect our analysis about the optical system under study, as in optical systems where the fibers have high birefringence [23,24].

From our results, a novel method was applied for the interferometric system analysis. The methodology was based on the relationship between the optical path difference and pole location, which was visualized in the pole-zero map. Our proposal finds potential applications on low-coherence interferometry systems, optical sensors, bridge visibility measurement, multiplexing systems based on interferometers, and optical source characterization. Therefore, our future work has two directions: practical applications and theoretical analysis. In the practical applications, signal processing and bridge visibility measurement can be implemented for optical fiber sensors. In the theoretical analysis, using the direct component permits us to obtain more information from the interference pattern, and this information could be studied on the complex s-plane.

## 5. Conclusions

In this work, using two unperturbed Fabry–Pérot interferometers, the cosine function filtered from both interference patterns, the Laplace transform, and a pole-zero map, we corroborated that pole location depends on the optical path difference. We also confirmed that the zero doesn’t give information about the interferometer because its location was over the origin of a pole-zero map. Both cases were theoretically and experimentally confirmed. In our analysis, the zero was over the origin, both poles were always at the imaginary axes, their locations depended on optical path differences, and each Fabry–Pérot interferometer generated two poles. In our experiments, the first Fabry–Pérot interferometer had the cavity length of ~1.6 mm, its optical path difference was of 2.33 mm, and the poles were localized at points ±i12. rad/nm. The second Fabry–Pérot interferometer had a cavity length of ~5.2 mm, its optical path difference was of 7.59 mm, and the poles were localized at the points ±i40.4 rad/nm. These experimental results confirmed our theoretical analysis.

Our proposal finds practical applications on coherence interferometers, signal processing for fiber optic sensors, and multiplexing systems based on the interferometry.

## Figures and Tables

**Figure 1 sensors-20-00453-f001:**
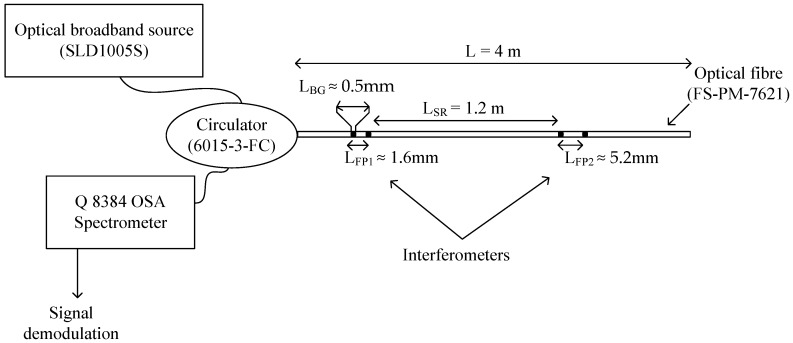
Optical system formed by two interferometers.

**Figure 2 sensors-20-00453-f002:**
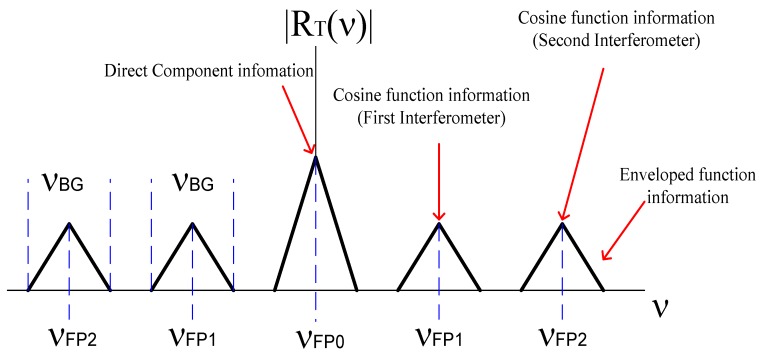
Optical signal formed by two interferometers.

**Figure 3 sensors-20-00453-f003:**
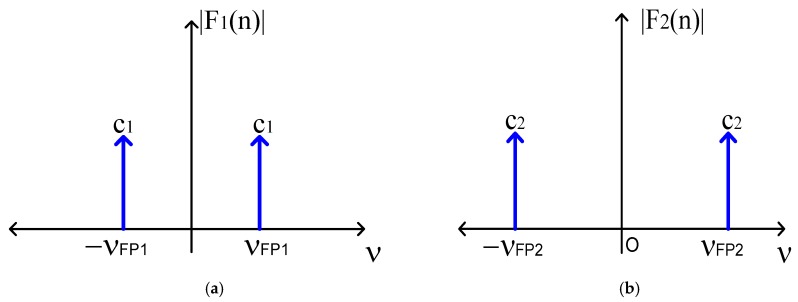
(**a**) cosine function $F_{1}\left(\nu \right)$ filtered from the first interference pattern; (**b**) cosine function $F_{2}\left(\nu \right)$ filtered from the second interference pattern.

**Figure 4 sensors-20-00453-f004:**
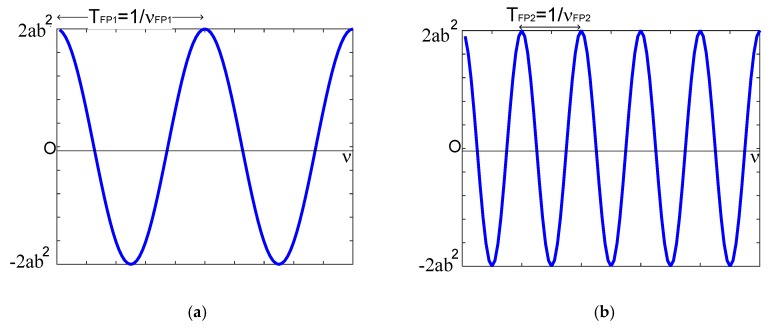
Cosine functions represented in wavelength domain: (**a**) function $f_{1}\left(\lambda \right)$; (**b**) function $f_{2}\left(\lambda \right)$.

**Figure 5 sensors-20-00453-f005:**
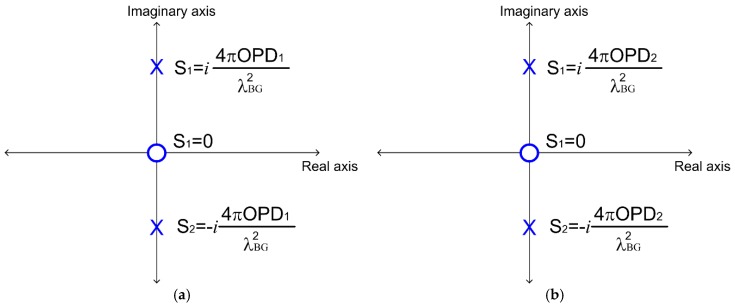
Pole-zero map obtained for the complex function $F_{m}\left(s\right)$: (**a**) the complex function $F_{1}\left(s\right)$; (**b**) the complex function $F_{2}\left(s\right)$.

**Figure 6 sensors-20-00453-f006:**
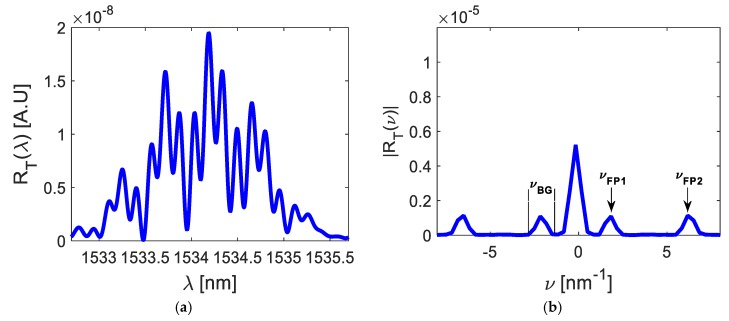
Optical signals measured: (**a**) the optical spectrum $R_{T}\left(\lambda \right)$; (**b**) the frequency spectrum $R_{T}\left(\nu \right)$.

**Figure 7 sensors-20-00453-f007:**
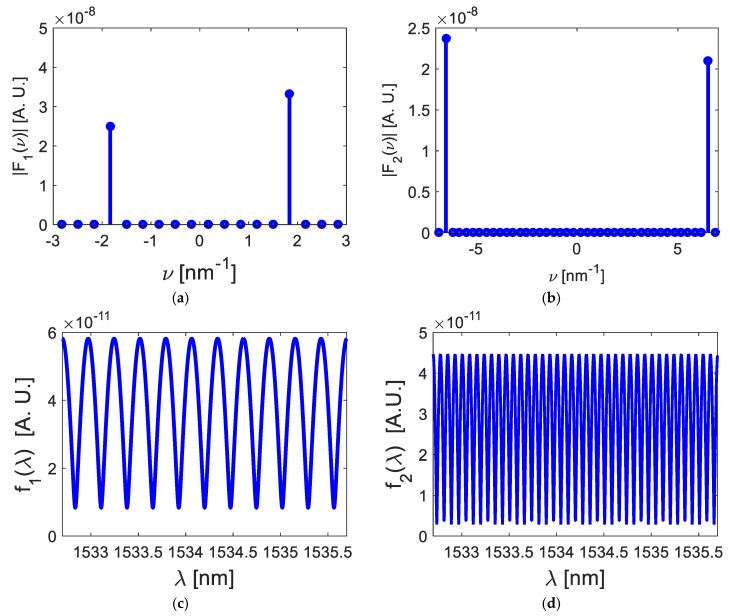
Cosine function representation: (**a**) $F_{1}\left(\nu \right)$ expressed in the frequency domain; (**b**) $F_{2}\left(\nu \right)$ expressed in the frequency domain; (**c**) $f_{1}\left(\lambda \right)$ expressed in the wavelength domain; (**d**) $f_{2}\left(\lambda \right)$ expressed in the wavelength domain.

**Figure 8 sensors-20-00453-f008:**
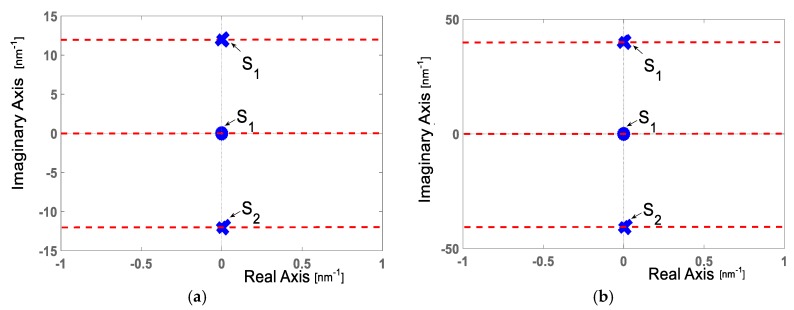
Pole-zero map representation: (**a**) complex function $F_{1}\left(s\right)$; (**b**) complex function $F_{2}\left(s\right)$.

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
