# Peer review of "Low-Finesse Fabry–Pérot Interferometers Applied in the Study of the Relation between the Optical Path Difference and Poles Location"

_sensors, 2020, doi:10.3390/s20020453_

Round 1

Reviewer 1 Report

The paper “Low-Finesse Fabry-Pérot Interferometers Applied in the Study of the Relation Between the Optical Path Difference and Poles Location” proposed a novel method to analyze the interference signal of multiple Fabry-Perot interferometers along a long fiber string. The paper first processed the interference signal with Fourier transform, then filtered it with Cosine function, and further conducted Laplace transform to obtain pole-zero map. The paper found that the locations of each pair poles are related with the optical path difference of each interferometer, and this can be used on analyzing interferometry signals with applications for optical fiber sensors and communication system. The paper is clearly written, and the experimental results are in accordance with the theoretical predictions. Hence I recommend its publication on Sensors.

Just one question, for the distance of L and Lsr in Figure 1 and Section 2.1, is the unit “mt” represents “meter”? If the Lsr is not as long as 1.2 meter, i.e., the two interferometers will have signal interference, will this method still be valid and match experiments well?

Author Response

Dear Reviewers

We appreciate your comments and questions, which improved our article. Following, we answer your comments and questions: Comments and questions are black and answers are blue.

Reviewer 1:

Just one question, for the distance of L and Lsr in Figure 1 and Section 2.1, is the unit “mt” represents “meter”? If the Lsr is not as long as 1.2 meter, i.e., the two interferometers will have signal interference, will this method still be valid and match experiments well?

Answers:

is the unit “mt” represents “meter”?

The term “mt” was changed for “m” in figure 1 and the text “section 2.1, line 85 and 87”

If the Lsr is not as long as 1.2 meter, i.e., the two interferometers will have signal interference, will this method still be valid and match experiments well?

If the LRS is not as long as 1.2 meter, both interferometers can produce an interference pattern and this interference can be detected by the OSA spectrometers. This method does not consider cross-talk noise, thus not being valid and match the experiments.  

Reviewer 2 Report

General comments:

In this paper, the authors propose optical fiber sensors, using Fabry-Perot interferomets in series to study of the relation between the optical path difference and poles location. Analyzing the manuscript the results don’t show this or the authors not show this clearly in the manuscript. The manuscript not response to several doubts describe in the following:

The introduction need to be improve. The authors needs clarify the problem and the possible solution. Why use the fiber FS-PM-7621. This fiber is polarization maintaining. The author not refer a study of effect of polarization in the results. the FBG is in this fiber? The symbol of meter is “m”. Change in the manuscript. The section 2.2 is based in the manuscript from reference 21 and 22. The authors is the same that this manuscript. This section can be removed. The information is similar to the 2 references and don’t increase quality. What is the difference between this manuscript and the references 21 and 22. Applications and experimental results for some physical parameter. The authors only change some mathematical approach. Is expect some applications. What is the advantage of this technique with others? What is the effect of temperature in the system? What is the influence of the cavity of each FPI and the separation between the 2 FPI in the results? The authors not explain. In section 3.3 the authors refer to poles location but in the manuscript this is confuse.

The manuscript is one part of the reference 21 and 22. The results not show an increase in the field, thus I recommend reject this manuscript.

Author Response

Dear Reviewer

We appreciate your comments and questions, which improved our article. Following, we answer your comments and questions: Comments and questions are black and answers are blue.

Reviewer 2:

Comments and Suggestions for Authors

The introduction need to be improve. The authors needs clarify the problem and the possible solution. Why use the fiber FS-PM-7621. This fiber is polarization maintaining. The author not refer a study of effect of polarization in the results. The FBG is in this fiber? The symbol of meter is “m”. Change in the manuscript. The section 2.2 is based in the manuscript from reference 21 and 22. The authors is the same that this manuscript. This section can be removed. The information is similar to the 2 references and don’t increase quality. What is the difference between this manuscript and the references 21 and 22. Applications and experimental results for some physical parameter. The authors only change some mathematical approach. Is expect some applications. What is the advantage of this technique with others? What is the effect of temperature in the system? What is the influence of the cavity of each FPI and the separation between the 2 FPI in the results? The authors not explain. In section 3.3 the authors refer to poles location but in the manuscript this is confuse.

Answers:

The introduction need to be improve.

The introduction was modified improving the explanation as can be seen in lines 74 to 79. This modification is done under your consideration.

The authors needs clarify the problem and the possible solution.

As mentioned in the later suggestion the problem and possible solution was also clarified as requested

Why use the fiber FS-PM-7621.

This fiber has low cost, good physical characteristics and the polarization is maintained. We don´t use a photosensitive optical fiber because this fiber has high cost.

This fiber is polarization maintaining.

Yes, the fiber is polarization maintaining. It can be read in the datasheet and we show you the screen printing, (Please, see pdf document).

The author not refer a study of effect of polarization in the results.

Your comment is very interesting, but our study does not consider polarization effects because the interferometers do not work by light polarization, as a consequence, polarization is not considered in the results, see references 11, 20-22. 

the FBG is in this fiber?

Yes, Bragg gratings are printed inside the fiber´s core. This is indicated in lines 72 and 73. However, we added the comment “inside the fiber´s core” into the section 2.1, paragraph 1 and line 91.

 The symbol of meter is “m”. Change in the manuscript.

The symbol of meter was changed to m where needed.

The section 2.2 is based in the manuscript from reference 21 and 22.

In reference 21, the quasi-distributed sensor based on frequency division multiplexing (FDM) and low-finesse Fabry-Perot interferometers was theoretically and numerically analyzed. From the analysis, limits and important physical parameters were determined. In reference 22, the Frequency-division multiplexing and wavelength-division multiplexing were combined increasing the number of local sensors. Both cases consider the maximum number of local sensors for the optical sensing system, as was described in the mathematical equations.

In our case, we consider a very particular situation regarding this quasi-distributed sensor in which the quasi-distributed sensor has only two low-finesse Fabry-Perot interferometers as Equation 1 describes.

Due to the previous comments we can say that the present work is based in the later work, but we considered using a different signal analysis technique.

The authors is the same that this manuscript.

We consider that the similarity of the different research papers mentioned does not make any conflict because in references 21 and 22 general cases are analyzed, while our article analyzes a particular case.

The three manuscripts are different. In each one of them the frequency response can be clearly differ as seen in their own frequency spectrum graphs and equation (figure 2 in the present work, figure 2 in reference 21 and equation 11 in reference 22).

This section can be removed.

We consider that the section should not be removed because the optical signal described through this equation is relevant for our proposed signal analysis.

The information is similar to the 2 references and don’t increase quality.

As mentioned in a later comment the two references are fundamental for our research

What is the difference between this manuscript and the references 21 and 22.

Our manuscript and references 21 and 22 are very different. In our manuscript, the optical signal is analyzed through the Fourier transform and Laplace transform. The Fourier transform is used to calculate the frequency spectrum and then both cosine functions are determined applying the inverse Fourier transform. Using these cosine functions, the complex function F(s) is determined, then we are able to calculate the poles and zeros which contain information about the optical path difference. The relation among them can be then depicted in the pole-zero map.  

On the other hand, in references 21 and 22 the optical signal is analyzed through the Fourier Domain Phases Analysis (FDPA) algorithm, which considers the enveloped and modulated functions of an interference pattern to calculate its shift. Additionally, for both references the instrumentation and sensors´ parameters, the optical system are considered for its numerical simulation.      

Based on the previous comments we consider that our manuscript and references 21 and 22 are different. Because we will propose a new methodology to analyze the optical signal obtained from the optical sensing system.

Applications and experimental results for some physical parameter.

In the present work, physical parameters are not considered in the objectives to be reached. Nevertheless, we are eager to consider physical parameters in future work as is indicated in section 4, paragraph 3, lines 250 -253.

The authors only change some mathematical approach. Is expect some applications.

Yes, we expect some applications as is indicated in future work, section 4, paragraph 3, lines 250 -253. Some examples are as a temperature sensor, strain sensor and others.

What is the advantage of this technique with others?

Your question is very outstanding. Although, we cannot mention some advantage because our proposal isn´t compared with some other signal processing techniques where the signal demodulation was done on optic sensors. However, the comparison is considered as future work, section 4, paragraph 3, lines 250 253

What is the effect of temperature in the system?

When the temperate affects interferometers, Bragg gratings and the optic fiber have elongation which affects the optical signal produced by both Fabry-Perot interferometers, making feasible to measure the temperature. In other words, Fabry-Perot interferometers can act as temperature sensors. It was demonstrated in reference 11.

What is the influence of the cavity of each FPI and the separation between the 2 FPI in the results? The authors not explain.

The cavity of each FPI is explained in the results section referring to the two FPI as F1 and F2 in the different equations from 27 to 34, and figure 8 for each one respectively.

The separation between the 2 FPI eliminates ghost interferometers for our sensing system. For this reason, it isn´t considered in the experimental results (See section 2.1, paragraph 1 and line 93).

 In section 3.3 the authors refer to poles location but in the manuscript this is confuse.

According to your comment we added some more detailed explanation on the poles location, in section 3.3 lines 213 and 214.

Round 2

Reviewer 2 Report

The authors presents  a version with some changes but not clarifying in full some doubts sent in the previous report. One is the polarization: if isn't important why use this fiber? Cheap is not justification because standard fiber is also cheap.

If this work is different from references 21 and 22 clarify this in the manuscript. Advantage and applications is important to understand the different options.

I recommend a major review where the authors needs clarify the previous questions.

Author Response

Dear Reviewer

We appreciate your comments and questions, which better our article. Following, we answer your comments and questions: Comments and questions are black and answers are blue.

Reviewer 2:

The introduction need to be improve. The authors needs clarify the problem and the possible solution. Why use the fiber FS-PM-7621. This fiber is polarization maintaining. The author not refer a study of effect of polarization in the results. the FBG is in this fiber? The symbol of meter is “m”. Change in the manuscript. The section 2.2 is based in the manuscript from reference 21 and 22. The authors is the same that this manuscript. This section can be removed. The information is similar to the 2 references and don’t increase quality. What is the difference between this manuscript and the references 21 and 22. Applications and experimental results for some physical parameter. The authors only change some mathematical approach. Is expect some applications. What is the advantage of this technique with others? What is the effect of temperature in the system? What is the influence of the cavity of each FPI and the separation between the 2 FPI in the results? The authors not explain. In section 3.3 the authors refer to poles location but in the manuscript this is confuse.

Answers:

The introduction need to be improve.

The introduction was modified improving the explanation as can be seen in lines 72-75 and 78-83. This modification is done under your consideration.

The authors needs clarify the problem and the possible solution.

Your comment is very significant. In the improved version of our manuscript, it can be verified that in lines 71 and 72 the problem arises to analyze “the dependence of the pole location due to the optical path difference”.

On the other hand, its possible solution is described in lines 72-87 along with numerical experimental results, which corroborate the solution of the problem stated in this work.

Why use the fiber FS-PM-7621.

We are very grateful for your observation at this point. In this research, we have considered using fiber FS-PM-7621 due to the following purposes:

The FS-PM-7621 fiber optic maintains the state of light polarization due to its low birefringence. This eliminates polarization effects in the optical system under study, such as those that occur in optical fibers with high birefringence (see reference 23 added to the article). As a consequence, our study on interferometric sensors is simplified, see references 11, 20-22. Comparing between a photosensitive optical fiber and fiber FS-PM-7621, for our possibilities, fiber FS-PM-7621 is much cheaper, easier to acquire, its physical characteristics (low attenuation and dispersion) are good because it operates in the third optical communications window, 1500-1600 nm and the Bragg gratings can be printed within that wavelength range. FS-PM-7621 fiber has low cost and is easy to acquire.

Note: Reference 23 was added to the article.

This fiber is polarization maintaining.

We are very grateful for your observation at this point. Based on FS-PM-7621 Datasheet (See Figure 1 in the file), yes, the fiber is polarization-maintaining. It simplifies our analysis about the optical system under study, see references 11, 20-23.

The author not refer a study of effect of polarization in the results.  

Your comment is very interesting. Due to the low birefringence of the FS-PM-7621 fiber, the polarization state of the light is maintained, which causes the polarization effects to be eliminated or not predominant in our optical system under study. For this reason, we do not consider the polarization effects in our results. See references 11 20-22.

However, point g, lines 250-252, was added in the discussion section.

In addition, references 23 and 24 were added.

the FBG is in this fiber?

Yes, Bragg gratings are printed inside the fiber´s core. This is indicated in line 76. Additionally, we added the comment “inside the fiber´s core” into the section 2.1, paragraph 1 and line 94.

 The symbol of meter is “m”. Change in the manuscript.

The symbol of meter was changed to m where needed.

The section 2.2 is based in the manuscript from reference 21 and 22.

We appreciate your observation made at this point. In this research work, section 2.2 is a particular case of references 21 and 22 since the Optical System under study consists of two Fabry-Perot interferometers, which are printed on the optical fiber. Therefore, it can be indicated that section 2.2 is based on the manuscript from references 21 and 22. However, it is convenient to consider the following points:

In reference 21, the optical system shown in Figure 1 was theoretically analyzed and numerically simulated. This Optical Sensing System is based on low-finesse Fabry-Perot interferometers and FDM. As a result of the analysis, its frequency domain multiplexing capacity (FDM) is determined, that is, the maximum number of Fabry-Perot sensors due to instrumentation, demodulation algorithm, optical sensor characteristics, and system noise. It is important to mention that, in this work, all the interference pattern information is used because the signal demodulation is done using the “Fourier Domain Phase Analysis” (FDPA) algorithm (see reference 11). In reference 22, the Optical Sensing System is theoretically analyzed and numerically simulated, using low-finesse Fabry-Perot interferometers and combining FDM / WDM techniques. Through the theoretical analysis, its multiplexing capacity was determined due to instrumentation, the demodulation algorithm, optical sensor characteristics, and system noise. Again, all interference pattern information was used because the "Fourier Domain Phase Analysis" (FDPA) algorithm (see reference 11) was applied in the demodulation of the optical signal. The present work is a particular case of references 21 and 22 because the Sensing System consists of two low-fines Fabry-Perot interferometers printed on an optical fiber but its Optical Path Difference (OPD) is studied in the complex plane s. The frequency of the interference patterns is represented on a pole-zero map and only the cosine function of the interference pattern is considered.

The authors is the same that this manuscript.

We are very grateful for your observation at this point. However, we consider the manuscripts not to be the same since their differences are observable in the equations and frequency figures. Please consider the following points:

In references 21 and 22, the maximum number of Fabry-Perot interferometers is indicated in the Optical System (multiplexing capacity), whereas, in our present work only two interference patterns are represented, see Equations 1 and 8. When plotting the frequency spectrum, the difference is more observable: in our work, there are only 5 frequency components (see Figure 2), one direct component, two positive components, and two negative components; in reference 21 there are 2M + 1 frequency components (see Figure 2), a direct component, M positive components and M negative components (See Figure 2, Reference 21); in reference 22 there are 2MK + 1 frequency components (see Equation 11, reference 22), a direct component, MK positive components, and MK negative components. In this case, there is an overlap of information in the frequency domain, see Figure 9 of the article: “Della Tamin M. and Meyer J. Quasidistributed Fabry-Perot Optical Fiber sensor for temperature Measurement. IEEE Access, 2018, 6, 66235-66242”.

This section can be removed.

Dear reviewer, we greatly appreciate your comments, which are undoubted to improve our work. However, in this case, the section cannot be removed because it is different from the previously reported cases and because the optical signal is the fundamental basis of our new analysis. This is clear by looking at the following sections of the manuscript.

The information is similar to the 2 references and don’t increase quality.

Dear reviewer, we greatly appreciate your comments to improve our work. In this case, we consider that the information is quite different between the two references (21 and 22) and our present work:

In references 21 and 22, the maximum number of Fabry-Perot interferometers is indicated in the Optical System (multiplexing capacity), whereas in our present work only two interference patterns are represented. When plotting the frequency spectrum, the difference is more observable: in our work, there are only 5 frequency components (see Figure 2), one direct component, two positive components, and two negative components; in reference 21 there are 2M + 1 frequency components (see Figure 2), a direct component, M positive components and M negative components (see Figure 2, Reference 21); in reference 22 there are 2MK + 1 frequency components (see Equation 11, Reference 22), a direct component, MK positive components, and MK negative components. In this case, there is an overlap of information in the frequency domain, see Figure 9 of the article: “Della Tamin M. and Meyer J. Quasidistributed Fabry-Perot Optical Fiber sensor for temperature Measurement. IEEE Access, 2018, 6, 66235-66242”.

The following three points describe more differences between this work and references 21 and 22.

In reference 21, the optical system shown in Figure 1 was theoretically analyzed and numerically simulated. This Optical Sensing System is based on low-finesse Fabry-Perot interferometers and FDM. As a result of the analysis, its frequency domain multiplexing capacity (FDM) is determined, that is, the maximum number of Fabry-Perot sensors due to instrumentation, demodulation algorithm, optical sensor characteristics, and system noise. It is important to mention that, in this work, all the interference pattern information is used because the signal demodulation is done using the “Fourier Domain Phase Analysis” (FDPA) algorithm (see reference 11). In reference 22, the Optical Sensing System is theoretically analyzed and numerically simulated, using low-finesse Fabry-Perot interferometers and combining FDM / WDM techniques. Through the theoretical analysis, its multiplexing capacity was determined due to instrumentation, the demodulation algorithm, optical sensor characteristics, and system noise. Again, all interference pattern information was used because the "Fourier Domain Phase Analysis" (FDPA) algorithm (see reference 11) was applied in the demodulation of the optical signal.

What is the difference between this manuscript and the references 21 and 22.

Dear reviewer, there is a big difference between our present work and references 21 and 22. In this work, we study theoretically, graphically (the pole-zero map) and experimentally the relationship between the Optical Path Difference (OPD) and the pole position using only the cosine function of the interference pattern. This new analysis will allow us to develop new graphic methods to study the dynamic behavior of interferometric systems. Whereas, in references 21 and 22 a theoretical analysis is made to determine the multiplexing capacity of the optical system considering the instrumentation, characteristics of interferometric sensors, signal processing and noise in the system.

Applications and experimental results for some physical parameter.

Dear reviewer, I greatly appreciate your comments to improve our work. As you know, the ultimate goal of all research is its application, being developed to solve a social problem. Unfortunately, we do not have a research laboratory on fiber optic sensors. We are working hard to implement one. At this time, we develop our work in academic laboratories. We only have a certain period for the implementation of the experiment and then it must be disassembled for academic reasons. This is the cause that prevents us from measuring physical parameters, but we consider for future work, section 4, paragraph 3, lines 254-261.

The authors only change some mathematical approach. Is expect some applications.

Dear reviewer, in this paper, we propose to analyze the relationship between the Optical Path Difference (OPD) and the pole-position, using only the cosine function of the interference pattern. Thus, not only some changes are made in mathematical expressions.

On the other hand, yes, we expect some applications as is indicated in future work, section 4, paragraph 3, lines 257-261. Some examples are temperature sensors, strain sensors, voltage sensors, oil detection, and others.

What is the advantage of this technique with others?

Dear reviewer, your question is very interesting. We consider that a graphic method applied to the dynamic analysis of interferometers would facilitate the study of interferometric systems since a graphic method is easier to interpret than an analytical method.

What is the effect of temperature in the system?

Dear reviewer, your question is very interesting. If the optical system is affected by temperature, the cavity and Bragg gratings will have some elongation and as a consequence, the reflection spectrum produced by the Fabry-Perot interferometers will have a shift that will be proportional to the resulting disturbance. Such a shift can be calculated with the FDPA algorithm and then the magnitude of the temperature can be measured, see reference 11.

That is, the optical system can be applied as a temperature sensor.

What is the influence of the cavity of each FPI and the separation between the 2 FPI in the results? The authors not explain.

As in the previous cases, your appreciation is very interesting.

Firstly, the separation between the 2 FPI eliminates ghost interferometers for our sensing system. For this reason, it is not considered in the experimental results (See section 2.1, paragraph 1 and line 96).

Secondly, because each Fabry-Perot interferometer has its own cavity, then each interferometer has its own OPD and as a consequence, each interference pattern has its own frequency. These frequency components are filtered using signal processing techniques, and then they are represented as a complex function F (s) with the Laplace transform, see Equation 32. Afterward, using Equation 32 the poles of each of the interference patterns (Equations 33 to 34) are determined and then its pole-zero map is plotted (see Figure 8). In the result plotted on the pole-zero map, it is graphically shown that for each cavity size or difference in the optical path, only one position on the pole-zero map corresponds.

 In section 3.3 the authors refer to poles location but in the manuscript this is confuse.

 Dear reviewer, we welcome your comments to improve our work. The process followed for the location poles is described below:

The theory presented in section 2 is described in the following points:

The optical signal of the Sensing System is detected using the OSA spectrometer and subsequently its frequency spectrum is calculated with the Fourier transform, section 2.2. The information of the cosine functions is identified in the frequency spectrum, filtered to eliminate the information of the envelope function of the interference pattern and finally its inverse Fourier transform is calculated to obtain a cosine function, which Contains the interference pattern frequency and Optical Path Difference (OPD) information, section 2.3. Applying the Laplace Transform to cosine functions, complex functions are determined. Such functions are used to calculate the location of poles and zeros of interference patterns (Equations 22 and 24) produced by Fabry-Perot interferometers, section 2.4. The poles and zeros are plotted on the pole-zero map (see Figure 5), section 2.4.

In the experimental results (section 3), the points of section 2 were proved:

Section 3.1, Figure 6a shows the optical spectrum detected with the OSA spectrometer and Figure 6b shows its frequency spectrum (Point 1). Section 3.2, Figures 7a and 7b show the cosine functions filtered from the frequency spectrum of Figure 6b. Whereas, Figures 7 c and 7 d show the cosine functions after applying the inverse Fourier transform (Point 2). Section 3.3, Equation 32 shows complex functions by applying the Laplace transform to cosine functions (Equations 28 and 29). Meanwhile, Equations 33 and 34 show the poles calculated for each interference pattern (Point 3). Section 3.3, Figure 8 shows the pole-zero map of the interference patterns of the optical system presented in Figure 1 (Point 4).

The presented description is basically the process described in our manuscript.

Round 3

Reviewer 2 Report

I agree with the changes made by authors. I recommend publish.